# Efficacy and Safety of Perampanel in Children with Drug-Resistant Focal-Onset Seizures: A Retrospective Review

**DOI:** 10.3390/children10061071

**Published:** 2023-06-17

**Authors:** Ahmed Elmardenly, Zekra Aljehani, Abdullah Tamim, Anas Alyazidi, Osama Muthaffar

**Affiliations:** 1Department of Pediatrics, King Faisal Specialist Hospital & Research Centre, Jeddah 23431, Saudi Arabia; aelmardenly@kfshrc.edu.sa (A.E.); atamim@kfshrc.edu.sa (A.T.); 2Medical/Critical Pharmacy Department, Pharmaceutical Care Division, King Faisal Specialist Hospital and Research Centre, Jeddah 23431, Saudi Arabia; zaljehani3@kfshrc.edu.sa; 3Department of Pediatrics, Faculty of Medicine, King Abdulaziz University, Jeddah 21589, Saudi Arabia; aalyazidi0015@stu.kau.edu.sa

**Keywords:** intractable seizures, drug-resistant seizures, pediatric, perampanel, efficacy, tolerability

## Abstract

Background: Epilepsy is one of the most common neurological disorders. Existing antiseizure medications (ASMs) are still unable to control seizures in one-third of these patients, making the discovery of antiseizure therapies with novel mechanisms of action a necessity. Aim of the Study: This study aimed to determine the safety and efficacy of perampanel (PER) as an adjuvant treatment for children with drug-resistant focal-onset seizures with or without focal to bilateral tonic-clonic seizures. Patients and methods: This is a single-center retrospective study of 38 epileptic pediatric patients, aged 2 to 14, at King Faisal Specialist Hospital and Research Center whose seizures were pharmaco-resistant to more than two antiseizure medications and followed for at least three months after PER adjuvant therapy initiation. Efficacy was assessed by the PER response rate at 3-, 6-, and 12-month follow-up evaluations, and side effects were also reported. Results: Multiple seizure types were reported. Myoclonic seizures were the predominant type of epilepsy in 17 children (44.7%). At 3 months, 6 months, and 12 months of follow-up, approximately 23.4%, 23.4%, and 18.4% of the patients were seizure-free at these time points, respectively. Adverse events were documented in 14 patients (35.7%) and led to the discontinuation of PER in 26.3%, 31.6%, and 36.8% of the studied group at the 3-, 6-, and 12-month follow-ups, respectively. The most common adverse events included dizziness or drowsiness, irritability, gait disturbance, and confusion; however, all were transient, and no serious adverse effects occurred. Conclusion: Our findings confirm the therapeutic efficacy of adjunctive PER in the treatment of drug-resistant epilepsy in children. As an adjunctive treatment for epilepsy, perampanel demonstrated sufficient effectiveness and tolerability.

## 1. Introduction

Epilepsy is a neurological disorder defined by at least two or more spontaneous seizures that happen more than 24 h apart [1]. Globally, 10% of the population experiences at least one seizure, with one-third developing epilepsy. The prevalence of epilepsy is age-dependent; it usually starts in childhood or even after the age of 60, whereas middle-aged people are less likely to develop it [2]. Epilepsy is a frequent neurological condition, with 50 to 70 cases per 100,000 people recorded annually [2]. In Saudi Arabia, the prevalence of active epilepsy is 6.5/1000 persons. Worldwide, the prevalence is similar, around 6.38/1000 persons [3]. In developing countries, there are variations in prevalence; for example, it is 3.9/1000 persons in India, 10.2/1000 persons in Tanzania, 13/1000 in Uganda, and 11/1000 persons in Bolivia. In other developed countries, the United Kingdom, for example, the prevalence is lower than in developing countries at 5.5/1000. In Norway, the prevalence is estimated at around 3.6/1000 persons, and in France, it is 5.4/1000 persons. These differences between countries could be explained by the different risk factors for epilepsy. In developing countries, the risk for labor complications, meningitis, and road traffic accidents is higher. The underdeveloped healthcare system also plays a role [4].

Drug-resistant epilepsy (DRE) is defined by the international league against epilepsy (ILAE) as “failure of adequate trials of two tolerated, appropriately chosen and used antiepileptic drug schedules (whether as monotherapies or in combination) to achieve sustained seizure freedom” [5]. Despite a myriad of available ASMs, 30–40% of people with epilepsy have seizures that cannot be controlled with the currently available drugs [4]. Children with drug-resistant epilepsy are more likely than adults to have cognitive problems, behavioral, and mental health problems, and a lower overall quality of life [5]. Consequently, there is an unmet need for novel pharmaceutical treatments that provide early and better seizure control and have fewer side effects.

Despite the rapid development of several antiseizure medications (ASMs), the percentage of patients who obtain optimal seizure control is unsatisfactory, and drug resistance rates remain high [6,7]. Pediatric patients with drug-resistant epilepsy have exceptional demand for seizure control. Understanding different mechanisms of epileptogenesis and drug resistance such as “drug transporter hypothesis, membrane expression of certain molecules, target theory, intrinsic severity model and others” will help in the invention of new drug therapies for epilepsy [8].

One of the most recent antiseizure medications, perampanel (PER), has a distinct mode of action and is the first-in-class, highly selective, non-competitive antagonist of the -amino-3-hydroxy-5-methyl-4-isoxazolepropionic acid (AMPA)-type glutamate receptor on postsynaptic neurons [9,10]. Excessive activation of glutamate receptors, including AMPA receptors, could trigger the development of seizures through modulation of epileptogenesis and overactivation of neuronal networks. By selectively inhibiting AMPA receptors, perampanel reduces excitatory neurotransmission and helps to prevent or reduce seizure activity [10,11]. In more than 40 countries worldwide, PER is approved as an adjunctive or monotherapy to treat focal-onset seizures with or without focal-to-bilateral tonic-clonic seizures in pediatric patients four years of age and older who have not responded to two or more antiseizure medications [10,11].

The effectiveness and tolerability of PER were proven in a phase III clinical program comprising three randomized, double-blind, placebo-controlled trials in patients aged 12 years or older with focal-onset seizures, with or without focal-to-bilateral tonic-clonic, who were receiving one to three ASMs [12,13,14,15]. In these studies, adjunctive PER treatment at daily doses of 4–12 mg was associated with significant decreases in the frequency of focal-onset seizures. Compared to placebo, analyses of pooled data from these trials revealed considerably higher response rates (>50% reduction in seizure frequency) [16]. A further round of randomized research demonstrated that adjunctive PER is effective and well-tolerated in adolescents with focal-onset seizures, with no clinically significant effect on behavior as judged by the Child Behavior Checklist [17].

However, there are insufficient clinical data on the tolerability and efficacy of PER in Saudi children (Middle Eastern/Arabic population) in clinical settings. Considering that race and ethnicity may account for changes in the pharmacokinetics and pharmacodynamics of medications, the response of PER in such patients should be carefully examined in terms of tolerability, effectiveness, and the associated differences in suggested doses. Moreover, PER’s novel mechanism of action, once-daily dosage, and minimal drug interactions are additional advantages of this add-on therapy. As a result, this study aims to determine the efficacy and tolerability of PER treatment in Saudi pediatric patients suffering from drug-resistant focal onset seizures with or without secondarily generalized seizures.

## 2. Materials and Methods

### 2.1. Patients

This is a single-center retrospective cohort study that was conducted on 38 pediatric patients with epilepsy aged 2 to 14 years who identified receiving PER as an adjunctive treatment for three months beginning on 21 February 2019 and ending on 29 September 2020 in the Pediatric Neurology Department of King Faisal Specialist Hospital and Research Center (KFSH & RC) in Jeddah, Saudi Arabia. KFSH & RC is a tertiary government referral center. The study reviewed patients between 21 February 2019 and 29 September 2020. RedCap (Vanderbilt University, Nashville, TN, USA) was utilized for electronic data collection. Only investigators had access to this database.

#### 2.1.1. Inclusion Criteria

The following were the criteria for patient enrollment: Pediatric patients aged 1 to 14 years who had epilepsy and had two or more antiseizure drugs fail (diagnosed with DRE). 

#### 2.1.2. Exclusion Criteria

The following were the criteria for exclusion: (i) adult patients > 14 years old; (ii) antiseizure naive patients; (iii) patients with primary generalized seizures; and (iv) patients with hepatic insufficiency or renal disorders. Ages above 14 to 18 years were not included because this is the policy in the hospital, after 14 years they follow with adult neurology.

### 2.2. Dose of Perampanel

Patients received PER at bedtime once daily in doses ranging from 2 mg to a maximum of 6–8 mg/day depending on the clinical response and tolerability, which are defined at the discretion of the treating neurologist. Baseline labs included hepatic and renal profiles. Follow-up in the pediatric neurology clinic should be at 3, 6, and 12 months. Treatment was discontinued when the neurologist determined that PER therapy was ineffective or when seizure exacerbation or unacceptable side effects were suspected.

### 2.3. Collected Data

The demographic data of the patient, laboratory data (liver and kidney function tests), age of onset of the disease, types of seizures, epileptic syndrome, duration of epilepsy prior to PER treatment, previous and concomitant antiseizure medications, highest PER dose, frequency of seizures at baseline and after treatment initiation, and reasons for the discontinuation of PER were extracted from medical records. Seizure types and epilepsy syndromes were classified based on criteria developed by the International League Against Epilepsy (ILAE).

The primary endpoint of efficacy data, the responder rate, was evaluated through seizure diaries given by the caretakers by comparing the frequency of seizures in the four weeks before PER application and after the maximal dosage of PER was achieved. The seizure frequency was extrapolated from medical records based on caregivers’ documentation. The responder rate was defined as the number of patients whose frequency of seizures in the last three months was reduced by >50% compared to the mean frequency of seizures at baseline.

Secondary endpoints include freedom of seizures, defined as no seizures during the previous three months. Tolerability was assessed through the documentation of possible adverse effects during treatment. Information on adverse events was recorded according to reports from patients, their parents, or their caregivers, or as recorded in the KFSHRC-J Quality Information System (QIS).

### 2.4. Sample Size Calculation

The sample size calculation was performed using G. power 3.1.9.2 (Universität Kiel, Kiel, Germany). The sample size was calculated according to responder rates in the adolescent patient groups, which were 4.8% for PER 2 mg (study 306), 23.1% for 4 mg (study 306), and 40.9% for 8 mg (all studies), according to a previous study [14]. Based on the following considerations, 0.05 α error and 80% power of the study, six cases were added to overcome dropout. Therefore, 38 patients will be allocated.

### 2.5. Data Analysis

Statistical analysis was conducted by SPSS v27 (IBM©, Armonk, IL, USA). A Shapiro–Wilk test and histograms were used to evaluate the normality of the distribution of data. Quantitative parametric data were presented as mean and standard deviation (SD) and were analyzed by an unpaired Student’s *t*-test. Quantitative non-parametric data were presented as the median and were analyzed by a Mann–Whitney test. The response of patients to PER therapy data on follow-up after 3, 6, and 12 months was assessed by repeated measures ANOVA which is the equivalent of the one-way ANOVA but for related, not independent, groups, and is the extension of the dependent *t*-test.

Qualitative variables were presented as frequency and percentage (%). The efficacy of PER was determined by comparing the frequency of all types of seizures in the month following PER initiation with the frequency of all types of seizures at the beginning of the study. The frequency of seizures was compared using Pearson’s chi-square and Fischer’s exact tests using odds ratios (OR). Logistic regression is also used to estimate the relationship between a dependent variable and one or more independent variables.

## 3. Results

### 3.1. Patients’ Demographics and Baseline Characteristics

Table 1 summarizes demographic information, characteristics of epileptic syndrome, type of seizures, and frequency before beginning PER treatment. The cohort comprised 24 males and 14 females who completed the study protocol, with mean ages of 8.32 ± 1.31 years (ranging from 2 to 13 years), and the majority of the studied group was in the 4–9 (44.7%) and 10–14 (47.4%) year old age groups. The average age of onset of the disease was 16.8 ± 23 months, ranging from one month to seven years. At baseline, all the studied population had normal liver function and only 2.6% had impaired kidney function. No changes in liver function tests or renal functions upon follow-up. Developmental delay was documented in 95% of the cohort.

Myoclonic seizures were the predominant type of epilepsy in 17 children (44.7%), followed by generalized tonic-clonic seizures (36.6%), where generalized seizures and drop attacks represented 23.7% of the study group. The mean monthly number of seizures among the studied cases at baseline was 92.6 ± 164.1 (range 2 to 900). Five children (15.2%) were diagnosed with Lennox–Gastaut syndrome, five with Dravet syndrome, one with autosomal dominant nocturnal frontal epilepsy, and the other 69.7% with other epileptic syndromes.

### 3.2. PER Dose

Table 2 shows that the average dose of PER is 5.6 ± 1.85 and 5.5 ± 1.81 (range: 2 to 8 mg) at 3- and 6-month follow-up, respectively, while the mean dose of additional PER at 12 months of follow-up is slightly higher at 5.8 ± 2.14 mg. The assessed seizure number was found to have decreased at the three-month follow-up (0–300)(5 patients) from the baseline (2–900)(30 patients). At the end of the follow-up, 24 patients (63.2%) continued with PER. At 3-, 6-, and 12-month follow-ups, respectively, around 26.3%, 31.6%, and 36.8% of the studied group discontinued therapy due to side effects or worsening of the condition (Table 2).

### 3.3. PER Efficacy

At 3-, 6- and 12-month follow-ups, approximately 23.4%, 23.4%, and 18.4% of the studied patients were seizure-free, whereas 15.2%, 15.8%, and 21.1% showed an overall response rate greater than 50% at 3-, 6-, and 12-month follow-ups, respectively. Among the cases, 15.8% demonstrated a lack of pharmacological improvement at the 3-month follow-up versus 10.5% at the 6-month follow-up and 5.3% at the 12-month follow-up (Table 2) (Figure 1).

Table 3 reveals that the responses to therapy and the positive therapeutic results are significantly correlated with younger age and earlier onset of the disease. Furthermore, 10% of responders had drop-attack-type seizures versus 41.2% of non-responders with a statistically significant difference, and 47.1% of non-responder cases had genetic epileptic syndromes versus 10% of responders with a statistically significant difference. There was a statistically significant improvement in the number of seizures at serial follow-up times among responders but not among non-responders.

### 3.4. Results of Tolerability

As illustrated in Figure 2, adverse events were reported in 24 patients (63.2%). The most frequent adverse events detected among the studied cases were dizziness and drowsiness (44.7%), followed by irritability and gait disturbance (23.7%); anxiety accounted for 13.2%, and confusion for 10.5%. Therapy had no adverse effects in 36.8% of the cases studied. Furthermore, no serious adverse effects were detected. All the side effects were reported during clinic follow-up. Around 26.3%, 31.6%, and 36.8% of the studied group at 3, 6, and 12 months, respectively, discontinued therapy due to side effects or worsening of the condition. No drug-to-drug interactions or allergic reactions were reported.

### 3.5. Logistic Regression

Variables presented in Table 4 were insignificant predictors for the response of patients to perampanel therapy.

## 4. Discussion

Perampanel (PER) has shown clinical importance as an adjunct medication in the treatment of elderly and adolescent patients with epilepsy. However, limited evidence was provided to assess the drug’s efficacy and reliability among pediatric patients with drug-resistant epilepsy. It was recently approved as a novel antiseizure drug for pediatric patients aged 4 and older with focal-onset seizures and individuals aged 12 and older with primary generalized tonic-clonic seizures. PER antagonizes alpha-amino-3-hydroxy-5-methyl-4-isoxazolepropionic acid (AMPA) in a non-competitive and selective manner [18]. AMPA is an excitatory synaptic transmitter, but if its receptor is overactivated, it may have an excitotoxic impact [19]. AMPA alterations can lead to a variety of diseases, most notably epilepsy. In this study, the effectiveness of PER in treating this problem was evaluated in a broad sample of pediatric patients.

In this study, the role of PER to address this condition was investigated among a large group of pediatric patients. The pooled data on the efficacy and safety of PER showed a lack of alteration to the hepatic profile. Furthermore, PER was similarly effective in controlling epileptic seizures in the short term (between three and six months) in approximately half of the study population. However, these percentages decreased to 18.4% during the year. Chang et al. [20] reported that approximately 13% and 10% of patients were seizure-free after 6 and 12 months of therapy for drug-resistant epilepsy in children, respectively. However, our results were consistent with previous research studies, which found seizure freedom ranging from 9 to 23% [21,22].

In this study, the number of responders was among the lowest in patients with Dravet syndrome (5%), contrary to findings in the literature which highlighted a good response to the treatment and a high efficacy that was maintained for more than a year [20] and a responder rate of up to 60% [21,23,24]. The lack of responders among Dravet syndrome can be interpreted in the context of the syndrome’s genetic background. Remarkably, recent investigations on Dravet syndrome estimated that 80% of the syndrome patients carry a mutation in the SCN1A gene [22,25,26] that encodes the voltage-gated sodium channel Nav1.1 [27]. Ca(2+)-permeable AMPA receptors are theorized to contribute to the process of epileptogenesis. Reduced GABA inhibition resulting from excitotoxic AMPA receptor-mediated processes may be associated with epilepsy, which may further impair GABA inhibition in Dravet syndrome [20,28].

Furthermore, a <50% seizure reduction was observed among 2 out of 38 (5.26%), which was dramatically less than other similar studies [29]. Seizure freedom was observed among <5% in the literature; however, in our cohort, the seizure freedom percentages were 23.68%, 13.15%, and 10.52% at 3, 6, and 12 months of follow-up, respectively [12,30]. Furthermore, the average dose of PER was between 2 and 8 mg at 3 and 6 months of follow-up and 5.8 ± 2.14 mg at 12 months. Subsequently, the discontinuation of medication increased significantly by 26.3%, 31.6%, and 36.8%, respectively. According to other research, a dose range of 4–12 mg/day can be administered without side effects or the discontinuation of therapy. Furthermore, although nearly one-third of patients reported no side effects from PER, most suffered episodes of dizziness or drowsiness. According to the research [23], adverse events are much more prevalent in patients aged 12 and older than those younger than 12 years of age.

Additionally, there were no significant differences in the efficacy between different types of seizure except in drop attack form. The response rate to the myoclonic form of seizure reached 40%. The non-responder rate was higher comparing a spectrum of genetic syndromes as illustrated in Table 3; however, a study that included 184 pediatric patients with refractory epilepsy receiving PER as an adjunctive treatment assured the safety of maintaining a low dose of PER with appropriate tolerability in children with refractory epilepsy of genetic etiology. The response rate for the myoclonic form of the seizure reached 40%, despite the fact that there was no significant variation in efficacy between other forms of seizure except for the drop attack form. When comparing a spectrum of genetic syndromes, as shown in Table 3, the non-responder rate was higher; however, a study involving 184 pediatric patients with refractory epilepsy receiving PER as an adjunctive treatment confirmed the safety and tolerability of maintaining a low dose of PER in children with genetic etiology refractory epilepsy. Furthermore, therapy cessation owing to adverse effects was most significant at 12 months. In a trial of 81 patients, adverse events were recorded in 47 (58%) patients taking PER, and 12 (15%) patients discontinued therapy due to adverse events. Yun et al. [31] supported our findings that dizziness was the most often reported side event. Regarding the incidence of adverse events, the two investigations had similar results [32].

The present study has some limitations. First, owing to the novelty of the study, the lack of previous studies nationwide made it difficult to compare the findings to others and among the same population. Another possible limitation is that this study was conducted with a relatively small sample size, which may have caused the possibility of a misconception. Additionally, recall bias may have played a role, as seizure frequency reporting was self-reported by patients and caregivers. Finally, due to the genetic background and pathophysiology of various syndromes that indicate poor outcomes, a variance in response rate and treatment efficacy can be detected in certain genetic syndromes.

## 5. Conclusions

This single-center retrospective cohort study offers additional evidence on the efficacy and safety of PER in the treatment of children and adolescents with drug-resistant epilepsy of various etiologies. This is the first study in the region, as far as we know, that reviewed the efficacy of PER in children with epilepsy. Based on the findings of this study and earlier research, PER was shown to be a successful and well-tolerated new therapeutic option for pediatric patients with pharmacoresistance to antiseizure medications.

## Figures and Tables

**Figure 1 children-10-01071-f001:**
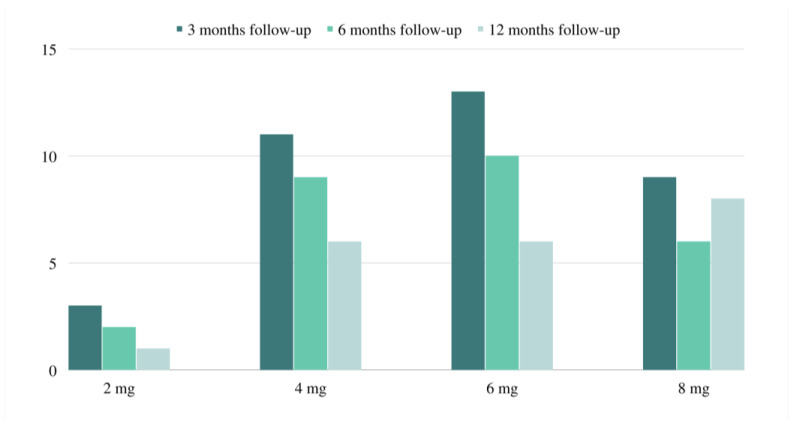
Dose of perampanel at 3, 6, and 12 months.

**Figure 2 children-10-01071-f002:**
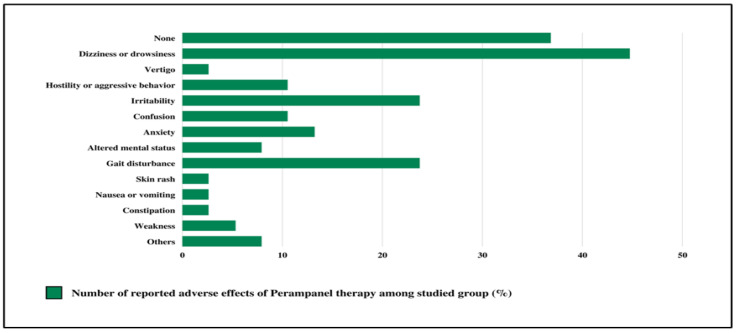
Adverse effects of perampanel therapy among studied group (in %).

**Table 1 children-10-01071-t001:** Baseline characteristics of the studied cases.

Age (years)	
Mean ± SD	8.32 ± 3.18
Range	2–13
Age groups (years)	
<4	3 (7.9%)
4–9	17 (44.7%)
10–14	18 (47.4%)
Sex	N (%)
Male	24 (63.2%)
Female	14 (36.8%)
Gestational Age	
Full term	30 (78.9%)
Preterm	2 (5.2%)
Unknown	6 (15.7%)
Gestational Age (weeks)	
Mean ± SD	39.4 ± 1.77
Birth weight (kg)	
Mean ± SD	2.9 ± 0.5
NICU Admission (*n*, %)	
Yes	10 (26.3%)
No	28 (73.6%)
Family history of seizures (*n*, %)	
Yes	11 (28.9%)
No	27 (71%)
Developmental delay	
Yes	36 (95%)
No	2 (5%)
Renal function test (RFT)	
Normal	37 (97.4%)
Impaired	1 (2.6%)
Liver function test (LFT)	
Normal	38 (100%)
Impaired	0
Alkaline phosphatase	
Mean ± SD	211.2 ± 154.6
Range	31–982 (IU/L)
Total bilirubin	
Mean ± SD	6.1 ± 13.1
Range	2–84 (µmol/L)
Age of disease onset (months)	
Mean ± SD	16.8 ± 23.3
Range	1–84
Type of seizure (patients)	
Focal without generalized seizure	2 (5.3%)
Focal with generalized seizure	9 (23.7%)
Generalized tonic-clonic	14 (36.8%)
Tonic	8 (21.1%)
Myoclonic	17 (44.7%)
Epileptic spasms	7 (18.4%)
Drop attacks	9 (23.7%)
Atypical absence	4 (10.5%)
History of status epilepticus	
Yes	19 (50%)
Epileptic syndrome	
Lennox–Gastaut syndrome	5 (15.2%)
Dravet syndrome	5 (15.2%)
Autosomal dominant nocturnal frontal epilepsy	1 (3%)
Other genetic syndromes	10 (30.3%)
Others	13 (39.4%)
Number of seizures at baseline	
Mean ± SD	92.6 ± 164.1
IQ Range (Median)	2–900 (30)
Previous ASMs (*n*)	(Median dose, IQR, mg/kg/d)
Carbamazepine (11)	19.5, (12–32.5)
Levetiracetam (28)	55, (42–83)
Valproic Acid (30)	31.7, (12–54)
Topiramate (26)	7, (3–11)
Lorazepam	0
Diazepam	0
Clobazam (25)	1.3, (0.4–2.2)
Clonazepam (10)	0.05, (0.04–0.1)
Phenytoin (3)	5.35, (4.7–6)
Rufinamide (3)	40, (30–40)
Gabapentin (1)	32
Lacosamide (5)	9, (3–10)
Lamotrigine (18)	4.3, (0.07–12)
Phenobarbital (9)	4, (2.8–5.3)
Oxcarbazepine (4)	21, (13–33)
Others	NA
Vigabatrin (3)	
Steripentol (1)	
Ethosuximide (3)	
EEG findings	
Epileptic encephalopathy	11 (29%)
Focal discharges	6 (15.7%)
Generalized discharges	12 (31.5%)
Slow background	9 (23.6%)
MRI brain findings	
Normal	13 (34%)
Atrophy	8 (21%)
Asphyxia	8 (21%)
Delayed myelination	6 (15.7%)
Hydrocephalus	3 (7.9%)

**Table 2 children-10-01071-t002:** Response of patients to perampanel therapy data on follow-up after 3, 6, and 12 months.

	At 3 Months N = 38	At 6 Months N = 38	At 12 Months N = 38	*p* Value
Dose of perampanel (mg)				
Mean ± SD	5.6 ± 1.85	5.5 ± 1.81	5.8 ± 2.14	P1 = 0.425
Range	2–8	2–8	1–8	P2 = 0.920
Number of seizures				
Mean ± SD	34.4 ± 61.2	33.2 ± 63.6	15.6 ± 23.2	P1 = 0.325
IQ Range (Median)	0–300 (5)	0–300 (5)	0–90 (4)	P2 = 0.246
Response to treatment				0.946
Discontinue therapy	10 (26.3%)	12 (31.6%)	14 (36.8%)	
Free seizures	9 (23.4%)	9 (23.4%)	7 (18.4%)	
>90%	5 (13.2%)	5 (13.2%)	5 (13.2%)	
>50%	5 (15.2%)	6 (15.8%)	8 (21.1%)	
<50%	3 (7.9%)	2 (5.3%)	2 (5.3%)	
No improvement	6 (15.8%)	4 (10.5%)	2 (5.3%)	
Therapy discontinuation				
Due to side effects	8 (21.1%)	10 (26.3%)	13 (35.1%)	0.905
Worsening condition	2 (5.3%)	2 (5.3%)	1 (2.6%)	

P1: *p*-value between patient response at 3 months and patient response at 6 months; P2 = *p*-value between patient response at 3 months and patient response at 12 months.

**Table 3 children-10-01071-t003:** Comparison of demographic and outcome data between responders and non-responders after 12 months follow-up.

	Responders N = 20	Non-Responders N = 18	*p*
Age (years)			
Mean ± SD	7 ± 3.5	9.81 ± 1.98	0.02
Range (Median)	2–12 (6)	6–13 (10)	S
Age groups			
<4	3 (15%)	0 (0.0%)	0.12
4–9	10 (50%)	7 (38.9%)	NS
10–14	7 (35%)	11 (61.1%)	
Sex			
Male	11 (55%)	13 (72.2%)	0.33
Female	9 (45%)	5 (27.8%)	NS
Renal function			
Normal	19 (95%)	17 (100%)	0.51
Impaired	1 (5%)	0 (0.0%)	NS
Age of disease onset (months)			
Mean ± SD	12.7 ± 19.7	20.6 ± 26.1	0.04
Range (Median)	1–60 (5)	1–84 (18)	S
Type of seizure	N = 20	N = 17	
Focal without generalized seizure	2 (10%)	0 (0.0%)	0.37 NS
Focal with generalized seizure	4 (20%)	5 (29.4%)	0.23 NS
Generalized tonic-clonic	5 (25%)	9 (52.8%)	0.08 NS
Tonic	3 (15%)	5 (29.4%)	0.29 NS
Myoclonic	8 (40%)	9 (52.8%)	0.43 NS
Epileptic spasms	4 (20%)	3 (15.8%)	0.54 NS
Drop attacks	2 (10%)	7 (41.2%)	0.03 S
Atypical absence	3 (15%)	1 (5.9%)	0.39 NS
Epileptic syndrome			
Lennox–Gastaut syndrome	2 (10%)	3 (15.8%)	0.28 NS
Dravet syndrome	1 (5%)	4 (21.7%)	0.12 NS
Autosomal dominant nocturnal frontal epilepsy	1 (5%)	0 (0.0%)	0.31 NS
Other genetic syndromes	2 (10%)	8 (47.1%)	0.01 S
Others	7 (35%)	6 (35.3%)	0.18 NS
Number of seizures at baseline (months)			
Mean ± SD	118.95 ± 210.2 *	61.6 ± 78.8 **	0.18
Range (Median)	2–900 (30)	4–270 (30	NS
Number of seizures at 3 m. follow-up (months)			
Mean ± SD	38.1 ± 76.3 *	29.5 ± 33.8 **	0.02
Range (Median)	0–300 (2.5)	0–120 (30)	S
Number of seizures at 6 m. follow-up (months)			
Mean ± SD	34.8 ± 72.5 *	28.7 ± 28.5 **	0.02
Range (Median)	0–300 (1)	1–90 (30)	S
Number of seizures at 12 m. follow-up (months)			
Mean ± SD	11.1 ± 16.8 *	34.5 ± 38.9 **	0.02
Range (Median)	0–60 (4)	4–90 (21)	S
Dose of perampanel\mg			
Mean ± SD	5.5 ± 2.1	5.6 ± 1.67	0.34
Range (Median)	2–8 (6)	2–8 (6)	NS

S: *p*-value < 0.05 is significant NS: *p*-value > 0.05 is not significant. * There was a statistically significant decrease in the number of seizures on serial follow-up among the responders’ group (repeated measures ANOVA test used). ** there was no statistically significant decrease in the number of seizures on serial follow-up responders, group showing improvement with therapy by more than 50%.

**Table 4 children-10-01071-t004:** Logistic regression for prediction of response of patients to Perampanel therapy.

	Coefficient	Std. Error	Odds Ratio	Wald	*p*
Age	−6.083	17,971.365	0.002	0.000	1.000
Sex	−9.752	41,614.802	0.000	0.000	1.000
Type of seizure	−6.508	16,503.681	0.002	0.000	1.000
Age of disease onset (months)	−0.733	1200.191	0.481	0.000	1.000
Dose of Perampanel at 3 months	16.067	38,884.962	9,500,000	0.000	1.000
Dose of Perampanel at 6 months	−27.386	51,467.596	0.000	0.000	1.000
Dose of Perampanel at 12 months	13.372	16,355.925	642,000	0.000	0.999
Number of seizures at baseline	0.015	534.797	1.015	0.000	1.000
Number of seizures at 3 m	0.996	4317.433	2.708	0.000	1.000
Number of seizures at 6 m	−1.129	3233.650	0.324	0.000	1.000
Number of seizures at 12 m	0.050	1039.211	1.051	0.000	1.000

## Data Availability

The datasets generated or analyzed during the current study are available from the corresponding author upon reasonable request.

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
