# Peer review of "Efficacy and Safety of Perampanel in Children with Drug-Resistant Focal-Onset Seizures: A Retrospective Review"

_children, 2023, doi:10.3390/children10061071_

Round 1

Reviewer 1 Report

Extensive English language editing required.

Author Response

Dear Reviewer,

Thank you for your valuable comments. We did discuss and review your comments point-by-point as you kindly suggested.

Kind regards,

Reviewer 2 Report

Comments:
This is a retrospective study which evaluates the efficacy of tolerability of Perampanel in children. The research question is not novel and many studies have been conducted on the same question in recent past. However, it is clinically important. Data on safety and efficacy from different parts of world with different ethnicity will help in assessing the clinical utility of the drug in true sense. I have following comments which needs to be addressed.
1-    What is the study design? Authors mention that the study was conducted between February 2019 till Sept 2020, this is the same period when the drug was administered. Hence the authors should clarify the design.
2-    The “redcap will have an electronic data…..”(line 97-98 ) needs to be rephrased.
3-    Inclusion criteria of “two drug fail” should be clearly defined and proper reference for the same should be provided.
4-    How the data were extracted? Who were involved in data extraction? Were they blinded? What measures were taken to address possible selection bias?
5-    The age range in inclusion criteria is 4-14 yrs, but in the table 1, age range is mentioned as 2-13. 3 patients were below 4 years of age. This anomaly should be corrected.
6-    Table 1: Units for AST/ALT/Alk phosphatase etc should be provided.
7-    Table 1: How can total bilirubin range be 2-84, and mean of 6.1 ?? While all patients had normal liver function.
8-    What is the unit of number of seizures in Table 1?
9-    What were the other ASMs used in the study cohort? What was the average number of ASMS used? How the authors addressed the effect of confounder of other ASMs in their analysis?
10-    The primary outcome was responder rate (i.e >50% reduction in seizure frequency from baseline). In table 2, under the column at 12 months, patients with  free seizures, >90% and >50% were 7,5 and 8 respectively. They add up to 20. Those <50% seizure reduction and no seizure reduction were 2 each which add up to 4.  So as per defined primary outcome of seizure responder rate,  non-responder number will be 4. So, I am not sure why the Non responder in Table is mentioned as 18? It seems the authors have considered those in whom the drug was discontinued (14) in non-responder category. Moreover, the therapy (PER) was discontinued in 13 patients due to side effects. Drug discontinuation due to side effect and treatment non-response are not synonymous. Hence, in my opinion the analysis in Table 3 is wrong.
11-    Authors may also refer to recent studies by Dan li et el (2022) on evaluating efficacy of PER in children. Ref:
-    Li D, Huang S, Wang X, Yang L, Song T. Efficacy and adverse reactions of perampanel in the treatment of epilepsy in children. Front Neurol. 2022 Jul 27;13:924057.
12-    Line 221, Cheng et al, the reference is wrong . Should be corrected .

Moderate editing of english language required.

Author Response

(The authors gave the same response as above.)

Round 2

Reviewer 1 Report

I would like to thank the authors for their replies. However, some points still need to be addressed. Some comments were already done but need further addressing.

However, there are some points that need to be addressed:

1)      The study design type should be present in the title.

2)      The abstract should be written in a single paragraph.

3)      The introduction still needs to be improved. For example, the authors could paraphrase the drug-mechanisms in their own words and add more information about them instead of adding the phrase about mechanisms  in  ”” .(Please see these articles for further reference: Janmohamed M, Brodie MJ, Kwan P. Pharmacoresistance–Epidemiology, mechanisms, and impact on epilepsy treatment. Neuropharmacology. 2020 May 15;168:107790.; Lerche H. Drug-resistant epilepsy—time to target mechanisms. Nature Reviews Neurology. 2020 Nov;16(11):595-6).

4)      The authors should provide more details about the study protocol and be more specific about the inclusion and exclusion criteria, the enrollment methods (how were patients recruited? Were they patients at an outpatient clinic?

We reviewed the charts of patients from Pediatric Neurology Clinic. All of them fulfilled the inclusion criteria and their clinical data were available. We aimed to make the inclusion criteria concise and specific for children with drug-resistant epilepsy taking PER as add-on treatment. We did include outpatient cases. Patients were recruited in data sheets in RedCap application.

I advise the authors to further detail the inclusion criteria of the study. The authors say children with DRE were included.  But, for example, in this reply to point 17) We did not count patients with intractable epilepsy (as it has no major relevance to the study and as the clinics has a wide mixture of epilepsy syndromes and that is time consuming to identify all of them).” So, according to this other reply it is not know which patients had DRE. Please further clarify this point and add your explanations to the text.

5)      About the setting was it a private or public setting, why this age cutoff was chosen, were people with other comorbid conditions excluded (which exactly?), was it digital record or not, which diagnostic criteria were used for diagnosing epilepsy, and drug-resistant epilepsy. Please add this information in the text.

King Faisal Specialist Hospital and Research Center is a tertiary governmental referral center.

The age from 4 years and above (inline with FDA approval). Below than 14 years (because this is the policy in our hospital, after 14 years they follow with adult neurology).

We excluded patients with hepatic insufficiency or renal disorders.

RedCap is an online, secure, electronic data collection tool.

We used ILAE definitions for epilepsy and drug-resistant epilepsy. We stated that under “collected data” section.

Epilepsy definition used: as a neurological disorder characterized by the following: At least two unprovoked (or reflex) seizures occurring greater than 24 hours apart. One unprovoked (or reflex) seizure and a probability of further seizures similar to the general recurrence risk (at least 60%) after two unprovoked seizures, occurring over the next 10 years.

Drug-resistant epilepsy definition used (stated in the introduction and in comment number 2 ): Drug-resistant epilepsy is defined by the international league against epilepsy (ILAE) as “failure of adequate trials of two tolerated, appropriately chosen and used antiepileptic drug schedules (whether as monotherapies or in combination) to achieve sustained seizure freedom.”

6)      The authors should work with a statistician to improve statistical analysis reporting. For example, there are analyses reported in the methods such as OR that are not mentioned in the results. The authors mentioned they performed repeated mesures ANOVA in the table legend, but the parameters are missing and ANOVA is not mentioned in the methods section.

We reviewed your comments with statisticians.

Response of patients to Perampanel therapy data on follow-up after 3, 6 and 12 months was assessed by repeated measured ANOVA which is Repeated measures ANOVA is the equivalent of the one-way ANOVA, but for related, not independent groups, and is the ex-tension of the dependent t-test as shown in Table 2.

Odds ratio (OR) is used within the Logistic regression analysis that is also used to estimate the relationship between a dependent variable and one or more independent variables as shown in Table 5.

Material and methods section has been modified. Tables also added and modified.

Parameters from statistical tests are still missing. Where are the confidence intervals in the logistic regression analysis? Here is a model of how repeated measures ANOVA should be reported, please work with a statistician for further improvements:

Also, the authors should analyze in the text each of the analyzes they performed. I cannot find in the text how the authors analyzed most of their statistical results. How did they interpret their logistic regression? In the discussion the authors should discuss their statistical results and compare to those from from the literature.

7)      Was data distribution normal? Were t-tests performed?

Not all data used was normally distributed, unpaired student t-test was used in Table 3 for Comparison of Dose of Perampanel (mg) between responders and non-responders after 12 months follow up.

I cannot find this table in the article. It is important to include it. Further details should be provided about these Perampanel dosages. Were these maintenance dosages? At which point were these doseages measured?

Responders

N=20

Non-responders

N=18

P

Dose of Perampanel (mg)

Mean ± SD

Range (Median)

5.5 ± 2.1

2-8 (6)

5.6 ± 1.67

2-8 (6)

0.34

NS

8)      Was the power of the study calculated?

What was the required sample size to have 80% power to detect effects in this study? Please provide further information about this point.

9)       How many patients had drug-resistant epilepsy? How many patients with DRE were given perampanel?

We did not count patients with intractable epilepsy (as it has no major relevance to the study and as the clinics has a wide mixture of epilepsy syndromes and that is time consuming to identify all of them).

38     patients with DRE were given PER.

This point needs to be further clarified since authors mentioned DRE was an inclusion criteria of the study.

10)   Please provide which were the AED the patients were previously using and at which dosage.

Added under table 1

Table should be significantly improved and reformatted.

19)   Was EEG performed?

Yes. Added under table 1

Table should be significantly improved and reformatted. Perhaps authors could create a separate table for diagnostic investigation details.

20)   Was neuroimaging performed?

Yes. Added under table 1

Table should be significantly improved and reformatted. Perhaps authors could create a separate table for diagnostic investigation details.

21)   Drug-response should be evaluated clinically but also through EEG and video-EEG. Were those methods used to evaluate drug response?

Yes. The drug response was evaluated clinically (seizure-response).

EEG follow up was not a major factor.

What about video-EEG? Was it performed to confirm DRE? How were Psychogenic Nonepileptic Seizure (PNES) excluded?

22)   What was the etiology of seizures?

Additional information was provided in table 1. As well as seizure classification and epilepsy syndrome types.

Gestational Age

Full term

30 (78.9%)

Preterm

2 (5.2%)

Unknown

6 (15.7%)

Gestational Age (weeks)

Mean ± SD

39.4 ± 1.77

Birth weight (kg)

Mean ± SD

2.9 ± 0.5

NICU Admission (n, %)

Yes

10 (26.3%)

No

28 (73.6%)

Family history of seizures (n, %)

Yes

11 (28.9%)

No

27 (71%)

Tables should be significantly improved and reformatted.

23)   How many had imaging abnormalities?

Added in table 1 (66% had imaging abnormalities).

Tables should be significantly improved and reformatted.

24)   In the tables, the authors should provide further clinical information such as follow-up time, whether or not there was failure of the first AED (which percentage), number of AED used and dosage, how many patients had multiple types of seizures/day, information about the frequency of seizures before diagnosis, mean seizure duration, family history of seizures, EEG abnormalities, neuroimaging abnormalities, developmental delay, comorbidities especially psychiatric but other types too, ethnicity of the patients, history of neonatal complications, history of status epilepticus and febrile seizures, gestational age and birth weight, school performance .

-Follow-up time: added

3, 6 and 12 months

-Whether or not there was failure of the first AED (which percentage): all patinets failed 1st medication and 2nd medications.

-Number of AED used and dosage: added in table 1

Previous ASMs (n)

Carbamazepine (11)

Levetiracetam (28)

Valproic Acid (30)

Topiramate (26)

Lorazepam

Diazepam

Clobazam (25)

Clonazepam (10)

Phenytoin (3)

Rufinamide (3)

Gabapentin (1)

Lacosamide (5)

Lamotrigine (18)

Phenobarbital (9)

Oxcarbazepine (4)

Others

Vigabatrin (3)

Steripentol (1)

Ethosuximide (3)

(Median dose, IQR, mg/kg/d)

19.5, (12-32.5)

55, (42-83)

31.7, (12-54)

7, (3-11)

0

0

1.3, (0.4-2.2)

0.05, (0.04-0.1)

5.35, (4.7-6)

40, (30-40)

32

9, (3-10)

4.3, (0.07-12)

4, (2.8-5.3)

21, (13-33)

NA

Tables should be significantly improved and reformatted.

-How many patients had multiple types of seizures/day: the number of seizure per day was difficult to extract from medical records

- Information about the frequency of seizures before diagnosis: discussed as percentage of improvement.

- Mean seizure duration: we could not get this info from patients records.

- Family history of seizures: added in table 1

All these comments should be addressed properly in the discussion as limitations of the study.

Family history of seizures (n, %)

Yes

11 (28.9%)

No

27 (71%)

Tables should be significantly improved and reformatted.

-EEG abnormalities, neuroimaging abnormalities: added

EEG findings

Epileptic encephalopathy

11 (29%)

Focal discharges

6 (15.7%)

Generalized discharges

12 (31.5%)

Slow background

9 (23.6%)

MRI brain findings

Normal

13 (34%)

Atrophy

8 (21%)

Asphyxia

8 (21%)

Delayed myelination

6 (15.7%)

Hydrocephalus

3 (7.9%)

- Developmental delay: added

Developmental delay

Yes

36 (95%)

No

2 (5%)

Tables should be significantly improved and reformatted.

Please add this information to the text/tables.

-Comorbidities especially psychiatric but other types too:

Some children had motor delay, autism spectrum, others have ADHD and cognitive delay. They were labelled as (developmental delay and added to the above table).

- Ethnicity of the patients: all are Arabic

- History of neonatal complications: added

Gestational Age

Full term

30 (78.9%)

Preterm

2 (5.2%)

Unknown

6 (15.7%)

Gestational Age (weeks)

Mean ± SD

39.4 ± 1.77

Birth weight (kg)

Mean ± SD

2.9 ± 0.5

NICU Admission (n, %)

Yes

10 (26.3%)

No

28 (73.6%)

Family history of seizures (n, %)

Yes

11 (28.9%)

No

27 (71%)

Tables should be significantly improved and reformatted.

-          History of status epilepticus and febrile seizures: added

Type of seizure

Focal without generalized seizure

Focal with generalized seizure

Generalized tonic clonic

Tonic

Myoclonic

Epileptic spasms

Drop attacks

Atypical absence

2 (5.3%)

9 (23.7%)

14 (36.8%)

8 (21.1%)

17 (44.7%)

7 (18.4%)

9 (23.7%)

4 (10.5%)

History of status epilepticus

Yes

19 (50%)

 Tables should be significantly improved and reformatted.

-          Gestational age and birth weight: added

Gestational Age (weeks)

Mean ± SD

39.4 ± 1.77

Birth weight (kg)

Mean ± SD

2.9 ± 0.5

Tables should be significantly improved and reformatted.

-School performance: not evaluated

25) Tables should be improved and unecessay informations such as AST and ALT levels excluded. Other tests such as Alkaline phosphatase, Renal function test (RFT), Liver function test (LFT), Total bilirubin. However, the authors should explain in the text that these tests were performed before starting the therapy and during the treatment to monitor drug tolerability and adverse effect.

ALT and AST were removed.

26) Table 2 needs to be restructured as it is not possible to read it as it is (configuration problems)

Table 2 corrected.

Table 2. Response of patients to Perampanel therapy data on follow-up after 3, 6 and 12 months.

At 3 months

N=38

At 6 months

N=38

At 12 months

N=38

P value

Dose of Perampanel (mg)

Mean ± SD

Range

5.6 ± 1.85

2-8

5.5 ± 1.81

2-8

5.8 ± 2.14

1-8

P1=0.425

P2=0.920

Number of seizures

Mean ± SD

IQ Range (Median)

34.4 ± 61.2

0-300 (5)

33.2 ± 63.6

0-300 (5)

15.6 ± 23.2

0-90 (4)

P1= 0.325

P2= 0.246

Response to treatment

Discontinue therapy

Free seizures

>90%

>50%

<50%

No improvement

10 (26.3%)

9 (23.4%)

5 (13.2%)

5 (15.2%)

3 (7.9%)

6 (15.8%)

12 (31.6%)

9 (23.4%)

5 (13.2%)

6 (15.8%)

2 (5.3%)

4 (10.5%)

14 (36.8%)

7 (18.4%)

5 (13.2%)

8 (21.1%)

2 (5.3%)

2 (5.3%)

0.946

Therapy discontinuation

Due to side effects

Worsening condition

8 (21.1%)

2 (5.3%)

10 (26.3%)

2 (5.3%)

13 (35.1%)

1 (2.6%)

0.905

P1: p value between patients response at 3 months and patients response at 6 months, P2= p value between patients response at 3 months and patients response at 12 months

These table is confusing as well as the multiple p values in some columns. Please work with a professional to improve statistical results reporting.

27) The authors could add graphs with PER dosages and remove dose values from the table to improve readability.

Figure 1 added

These graph is out of scale. Please work with a professional to improve statistical results reporting.

28) The authors should be more specific about how was seizure frequency was calculated and how response to treatment was evaluated.

The seizure frequency was extrapolated from medical records based on caregivers’ documentation. We agree that it would be ideal to have a unified documentation method, however, in such retrospective review we found it difficult to call families again to recap and discuss past and present seizure frequencies and response. We found that “percentage” was documented in their files and we agree it’s a subjective evaluation.

These limitations should be added in the text.

29) The authors should provide more details about treatment course such as how were adverse effects managed, drug-interactions, allergic reactions, rate of patients that presented low adhesion.

All the side effects were reported during clinic follow up. around 26.3%, 31.6%, and 36.8% of the studied group at 3, 6, and 12 months respectively, discontinued therapy due to side effects or worsening of the condition. No drug-to-drug interactions or allergic reactions were reported. We did not comment of adherence to PER as it was not reported (i.e. good compliance).

All of these details should be disclosed in the text.

30) The graph in figure 1 should have its scale adjusted.

We  added a comment (percentage)(renamed into figure 2)

These graph is out of scale. Please work with a professional to improve statistical results reporting.

31) Was there any difference in adverse effects between patients with and without DRE?

All our patients were DRE. PER was added in DRE.

Further clarify this point as the previous comments above asked.

32) Tables 1 and 3 have repetitive information. I advise authors to refer to other articles in the literature similar to their to improve their tables, methods and results reporting. (Please see these for further reference: Vecht, C., Duran-Peña, A., Houillier, C. et al. Seizure response to perampanel in drug-resistant epilepsy with gliomas: early observations. J Neurooncol 133, 603–607 (2017). https://doi.org/10.1007/s11060-017-2473-1; Chang FM, Fan PC, Weng WC, Chang CH, Lee WT. The efficacy of perampanel in young children with drug-resistant epilepsy. Seizure. 2020 Feb 1;75:82-6, Huber B, Schmid G. A two-year retrospective evaluation of perampanel in patients with highly drug-resistant epilepsy and cognitive impairment. Epilepsy & Behavior. 2017 Jan 1;66:74-9.)

Thanks. Modified and Chang et al is one of our references.

Tables should be significantly improved and reformatted.

 33) Especially because the authors have stressed they treated patients with myoclonic epilepsy with PER, their EEG data should be disclosed.

EEG table added.

Tables should be significantly improved and reformatted.

34) Discussion needs to be completely restructured and authors should compare their data to other relevant published articles and stress what does their study add new, and add further details about limitations of the study.

We added and modified on the discussion part (highlighted) and limitations added.

The authors should analyze in the text each of the analyzes they performed.I cannot find in the text how the authors analyzed most of their statistical results. How did they interpret their logistic regression? In the discussion the authors should discuss their statistical results and compare to those from from the literature. Please add further details about limitations as discussed in this review.

Minor editing required.

Author Response

(The authors gave the same response as above.)

Reviewer 2 Report

Comments:

The authors have addressed most of my queries to previous revision. However, some points still need to be addressed:

1-     The “ two drug fail” (Reply to query 3) criteria of ILAE should be in the inclusion criteria not in the introduction part.

2-     The inclusion criteria still mentions the age to be 4-14years (Line 143-144)

3-     Line 153, ‘follow-up……..’the line should be in past sense.

4-     Authors have added a sample size calculation section in the new submission. However, I find this more confusing, misleading and erroneous. Firstly, the figures of the response rates mentioned for different doses in the referenced studies are not what has been mentioned in those articles. Secondly, why loss to follow up in a retrospective study?? The outcomes have already happened. If one has inadequate data, they can be excluded. Keeping these points aside, the most important point to be considered, is the sample size calculation as mentioned by the authors valid for the study design adopted? Authors may refer to following materials for determining how to calculate sample size for a retrospective chart review.

-Gearing RE, Mian IA, Barber J, Ickowicz A. A methodology for conducting retrospective chart review research in child and adolescent psychiatry. J Can Acad Child Adolesc Psychiatry. 2006 Aug;15(3):126-34.

-Sackett DL, Haynes RB, Tugwell P. Clinical epidemiology: a basic science for clinical medicine. Little, Brown and Company; 1985.

5-     Moreover, in retrospective chart review the methods of sampling has to be described. For example: Convenience, quota and systemic.  Which one was applied in this study should be mentioned.

6-     Response to query no 7: I think the authors did not understand what I meant. The total bilirubin has been mentioned as 6.1 ±13.1 (mg/dl)  (Range 2-84). But the normal range of Total bilirubin is 0.1 -1.2 mg/dl. So all patients in this study cohort had jaundice??

7-     The dosage for the other ASMs mentioned in the Table 1 should be rechecked. The median dose and IQR in mg/kg/d has been provided. The upper limit in the IQR for some AEDs exceeds the permissible limit.

8-     In table 2, the p value of 0.946 corresponds to which variables? Is it overall or to >90%?

9-     To my previous query no 10: To club non-responders (4) to drug discontinuation group (14) for statistical purpose will give an erroneous conclusion. Authors have to state that at which time (6 month or 12 months) , they intend to evaluate the response to therapy (>50% improvement). If intended to be done at 12 months, the analysis can be done  with 20 (responders) vs 6 ( 4 -non-responders + 2 discontinuation due to worsening condition (if that means worsening seizures). They may provide the reason for same in the footnote or discussion part.  

10- Table 4 and Figure 2 convey the same information. Only one of them may be provided.

11- For my query no 1, If I am not mistaken, the follow up data were also collected from the database.  The follow up of 1 year, was also completed by September 2020 ?. If so, all included patients were given the drugs from Feb 2019 till September 2019 and then 1 year follow up was done? Or the patients were selected from Feb 2019 till September 2020, and then 1 year follow up was done till September 2021? This should be clarified

12- The newly added reference 33, should be mentioned in the manuscript also.

Minor editing of english language required.

Author Response

(The authors gave the same response as above.)
